# Online Mixed Missing Value Imputation Using Gaussian Copula

Eric Landgrebe [1]   Yuxuan Zhao [1]   Madeleine Udell [1]

## Abstract

Many data science algorithms require complete observations, making missing value imputation an important step in many data processing pipelines. Imputation is also of independent interest for applications such as recommender systems. To address real-world big data problems, imputation algorithms must handle mixed data, containing ordinal, boolean, and continuous variables, and such algorithms must be highly scalable. In this work we develop a semi-parametric online algorithm for mixed missing value imputation using a Gaussian Copula. This online algorithm improves on the speed of its offline counterpart by an order of magnitude, with similar accuracy. The online method can also improve on the offline method by adapting to a changing data distribution.

## 1. Introduction

Many modern datasets are messy, containing missing values and a mixture of ordinal, binary, and continuous data. These datasets, such as those containing survey or rating data, may also be massive, with many millions of data points. Many modern machine learning algorithms require completely observed data; hence imputation of the missing entries is an important preprocessing step. A Gaussian copula imputation model has recently shown promising results on a variety of moderately sized mixed datasets (Zhao & Udell, 2019). This model posits that the data is generated by drawing a latent vector from a multivariate Gaussian with zero mean and unit variance that is then scaled by an elementwise monotonic function to match the marginal distributions of each observed feature. This is a natural model for mixed data, as the model posits that ordinals are the result of thresholding a continuous latent variable. In the case of say, product rating data, we can imagine the actual ordinal values as being the result of rounding some continuous affinity for a given product. The learned correlation matrix of the latent multivariate Gaussian can be used for imputation or to correlate features.

**Online Imputation**  Online data, generated by sensor networks, financial transactions, or ongoing surveys, present a substantial challenge for efficient data analysis. These datasets could also contain mixed and highly missing data, as sensors fail intermittently, or survey respondents fail to respond. In this setting, we make sequential observations, with missing values, and must impute the missing values and suffer the associated error before we can see future data.

**Contribution**  This paper develops an online algorithm for missing value imputation using the Gaussian copula model that incrementally updates model parameters, imputing based only on past, not future data. Hence this method can adapt to a changing data distribution. We also develop a mini-batch offline algorithm. Both algorithms are faster than previous methods while maintaining accuracy.

## 2. Related Work

**Offline**  One class of offline methods for data imputation uses generalized low-rank models (Udell et al., 2016), which include Principal Component Analysis (PCA) as well its generalizations. Broadly speaking, this class of algorithms minimizes the misfit of the data to a low rank parameter matrix, together with regularizers that impose a desired structure. XPCA by Anderson-Bergman et al. (2018) generalizes the idea of PCA to account for mixed data. MissForest (Stekhoven & Bühlmann, 2011) is a non-parametric method that imputes missing data using random forests. It iteratively updates estimates of missing values in each column with random forests trained on the features in the other columns.

**Gaussian Copula**  Most directly related to this work is Zhao & Udell (2019), which trains a Gaussian copula using an approximate expectation maximization (EM) algorithm in the offline setting. Our work generalizes this algorithm to the online setting by using an online EM algorithm to fit the copula and an online algorithm to estimate the marginals.

**Online**  GROUSE (Balzano et al., 2010) is a popular algorithm for online low-rank matrix completion and subspace

---

[1]Cornell University. Correspondence to: Eric Landgrebe <ecl93@cornell.edu>, Yuxuan Zhao <yz2295@cornell.edu>, Madeleine Udell <udell@cornell.edu>.

*Presented at the first Workshop on the Art of Learning with Missing Values (Artemiss) hosted by the 37$^{th}$ International Conference on Machine Learning (ICML).* Copyright 2020 by the author(s).

tracking using gradient descent on the Grassmannian. Jin et al. (2016) give a provable online method for low-rank matrix completion using gradient descent on a non-convex objective function. These methods both assume the data is low rank and real-valued. In contrast, our method handles high-rank, mixed data as well. Other authors have proposed online high-rank matrix completion methods that exploit low-dimensional manifold structure in the data (Fan & Udell, 2020), but these methods do not target mixed data.

**Our Contribution: Online Copula Estimation** The main contribution of this work is to adapt of the approximate EM algorithm of Zhao & Udell (2019) to the online setting. We formulate this online variation by applying ideas from Cappé & Moulines (2009) to the EM formulation of Zhao & Udell (2019) to estimate the copula correlation matrix online. We also develop a parallel Python implementation of the algorithm of Zhao & Udell (2019) and a version that uses minibatches of samples. We show that our minibatch fitting algorithm improves speed while maintaining accuracy compared to Zhao & Udell (2019). We also show empirically that our online algorithm can adapt to a changing data distribution.

## 3. Methodology

**Notation** We use capital letters to denote matrices and lower-case letters to denote vectors. Data matrices will have columns representing features and rows representing examples. We use $X_j$ to denote the $j$th column of matrix $X$. We use $x^i$ to denote the $i$th row of $X$, and $x_i$ to denote the $i$th entry of a vector $x$. We use $x^i_{\mathcal{O}_i}$ to denote the observed entries (both continuous and discrete) of row $i$ of matrix $X$. We denote by $\mathcal{E}$ the elliptope (the set of all correlation matrices). All proofs and many details appear in the appendix.

**Fundamentals** Given an elementwise monotone function $f$ and correlation matrix $\Sigma$, we say that a random variable $x \in \mathbb{R}^d$ is drawn from the Gaussian copula $x \sim GC(f, \Sigma)$ if $x = f(z)$ with $z \sim N(0, \Sigma)$. In other words, we generate a random variable $x$ from the Gaussian Copula by first drawing a latent vector $z$ from a normal distribution with mean 0 and correlation $\Sigma$, and then applying the elementwise monotone function $f$ to $z$ to produce $x$. It is easy to show that the monotone function $f$ for a given copula is unique (Zhao & Udell, 2019): indeed, if the cumulative distribution function (CDF) for $X_j$ is given by $F_j$, then $f_j = F_j \circ \Phi^{-1}$ where $\Phi$ is the standard Gaussian CDF. We note that for ordinal values, the CDF is a step function, so $f_j^{-1}$ is set-valued.

**Imputation** It is straightforward to impute missing values given a copula $f$ and correlation $\Sigma$ by applying the marginal transformation $f$ to the conditional mean of the latent $z$ conditioned on the observations. Zhao & Udell (2019) provide

an explicit formula for this conditional mean.

**Marginal Estimation** Given an estimate of the CDF $\hat{F}_j$ of the $j$th column, we can estimate the marginals $f_j$ as $\hat{F}_j \circ \Phi^{-1}$. In the offline setting, it is natural to use the empirical CDF. In the online setting, we replace this with an online estimate to the CDF over data observed so far.

### 3.1. Warmup: Offline Expectation Maximization

In this section we review a maximum likelihood estimation for the correlation of our copula $\Sigma$ under mixed observations in the offline setting as developed by Zhao & Udell (2019).

**Completely Observed Continuous** First, if the data matrix $X$ were fully observed with continuous values, we could compute the latent variable $z^i = f^{-1}(x^i)$ for each row $i$. Maximum likelihood estimation (MLE) in this setting reduces to MLE of a multivariate Gaussian. The log likelihood of the observed data matrix for a given correlation $\Sigma$ is

$$\ell(\Sigma; X) = c - \frac{\log(|\Sigma|)}{2} - \frac{1}{2} \operatorname{Tr}(\Sigma^{-1} \sum_{i=1}^n \frac{1}{n} z^i (z^i)^\top),$$

maximized by the empirical covariance $\frac{1}{n} \sum_{i=1}^n z^i (z^i)^\top$.

**Mixed Partially Observed** More generally, given data matrix $X$ with partially observed rows with mixed values, MLE maximizes the observed likelihood. Denote by $\phi(\cdot; 0, \Sigma)$ the PDF of a normal vector with 0 mean and correlation $\Sigma$. We seek to maximize (over $\Sigma$) the observed likelihood, integrating over all latent variables $z^i \in f^{-1}(x^i)$ that map to the observed values $x^i$. We note that $f^{-1}$ maps missing values to $\mathbb{R}$, so $f^{-1}(x^i)$ is fully determined by $x^i_{\mathcal{O}_i}$ and the observed log likelihood is:

$$\ell(\Sigma; \{x_{\mathcal{O}_i}\}_{i=1}^n) = \frac{1}{n} \sum_{i=1}^n \log \left( \int_{z^i \in f^{-1}(x^i)} \phi(z^i; 0, \Sigma) dz^i \right).$$

This integral is difficult to compute and even more difficult to optimize. Zhao & Udell (2019) propose an EM algorithm to avoid this difficulty. At each iteration, the EM algorithm updates the covariance estimate $\Sigma^t$ as

$$\Sigma^{t+1} = \sum_{i=1}^n \frac{1}{n} \mathbb{E}[z^i (z^i)^\top | x^i_{\mathcal{O}_i}, \Sigma^t].$$

This update is easy to understand: we estimate covariance by an "empirical covariance matrix" of latent variables $z^i$. The expectation weights these $z^i$ by their likelihood given the observations $x^i_{\mathcal{O}_i}$ and the previous covariance estimate $\Sigma^t$. We approximate these expectations as in Zhao & Udell (2019) using the fact that $z^i | x^i_{\mathcal{O}_i}, \Sigma^t$ is a truncated normal variable. Finally, we ensure the covariance matrix has a unit diagonal using a diagonal scaling; see Zhao & Udell (2019).

## 3.2. Generalization to the Online Setting

The key insight of Cappé & Moulines (2009) is to replace the expectation step of an EM algorithm with an incremental update to a previous estimate of the expectation. This approach yields an online algorithm: we simply update our expectation estimate as data comes in, and do not need to retain all data to perform an expectation update. The maximization step in this framework is unchanged. Formally, Cappé & Moulines (2009) propose to update the likelihood $Q$ with the rows $S \subseteq \{1, \ldots, n\}$ observed at time $t$ as

$$Q_{t+1}(\Sigma) = (1 - \gamma_t)Q_t(\Sigma) + \gamma_t Q(\Sigma; \Sigma^t, X_S^t) \quad (1)$$

with a monotonically decreasing stepsize $\gamma_t \in (0, 1]$.

The EM algorithm for the Gaussian copula takes a particularly simple form: maximizing Eq. (1) wrt $\Sigma$, we find

$$\Sigma^{t+1} = (1 - \gamma_t)\Sigma^t + \gamma_t \frac{1}{|S|} \sum_{i \in S} \mathbb{E}[z^i(z^i)^\top | x_{\mathcal{O}_i}^i, \Sigma^t]. \quad (2)$$

We then scale the correlation as in the offline setting. Each expectation $\mathbb{E}[z^i(z^i)^\top | x_{\mathcal{O}_i}^i, \Sigma^t]$ can be computed in time $O(\alpha p^3)$, where $\alpha$ is the fraction of missing entries. The computation also parallelizes over rows $i \in S$.

# 4. Experimental Results

**Overview and Metrics** We compare the results of our algorithm, with and without parallelism and batching, to those of missForest (implemented in missingpy), a state-of-the-art algorithm for mixed imputation using random forests, limited to 10 iterations. We use variable thresholds and maximum iterations for the EM algorithm, and report the runtime in lieu of reporting all parameters. Code for our algorithms and experiments is available at https://github.com/udellgroup/online_mixed_gc_imp.

In what follows, "Standard EM" denotes the EM algorithm of Zhao & Udell (2019); "Online EM" denotes a fully online version of the EM algorithm, which estimates marginals online and fits the covariance using Eq. (2) with rows $S$ chosen sequentially; and "Minibatch EM" denotes a version of the EM algorithm that uses offline marginal estimates and fits the covariance using Eq. (2) but with the rows $S$ chosen according to a random permutation of the dataset. Finally, "threaded" denotes the use of 2 cores and 2 additional virtual cores for parallelism to compute the expectations.

Our tests measure the mean absolute error (MAE), root mean squared error (RMSE), and scaled mean absolute error (SMAE), defined as SMAE $\equiv |\hat{X}_j - X_j|/|X_j^{\mathrm{med}} - X_j|$ where $\hat{X}_j$ are the imputed values of column $i$ and $X_j$ are the true values, and $X_j^{\mathrm{med}}$ is the median value in column $j$, all restricted to the masked entries in column $j$. Hence median imputation has SMAE $= 1$, and an algorithm that imputes better than the median has SMAE $< 1$.

All experiments use a 2015 macbook pro with a 2.7 GHz Intel Core i5 processor and 8 GB RAM. All threading uses 2 cores and 2 additional virtual cores. All implementations use python to allow for a direct comparison of runtimes.

## 4.1. Generated Data

**Dataset** We first consider a synthetic dataset consisting of 15-dimensional vectors drawn from a Gaussian Copula, with 5 continuous, 5 ordinal with five levels, and 5 binary entries. We mask 30 percent of entries as a test set for evaluation and report mean results over ten repetitions, for a standard, minibatch, and online variant of the Gaussian copula EM algorithm. We report runtime, RMSE, and SMAE for continuous (cont), binary (bin), and ordinal (ord) values.

*Table 1.* Runtimes and Errors on Simulated Copula Dataset.

| Method | Threads | Runtime (s) | SMAE (Cont) | SMAE (Bin) | SMAE (Ord) | RMSE |
|--------|---------|---------|--------|-------|-------|------|
| Standard EM | no | 90.422 | 0.774 | 0.674 | 0.787 | 0.316 |
| Standard EM | yes | 39.802 | 0.774 | 0.674 | 0.787 | 0.316 |
| Minibatch EM | no | 38.849 | 0.773 | 0.673 | 0.785 | 0.316 |
| Minibatch EM | yes | 19.856 | 0.773 | 0.673 | 0.785 | 0.316 |
| Online EM | yes | 14.792 | 0.8 | 0.704 | 0.82 | 0.333 |
| Missforest | no | 65.184 | 0.945 | 0.743 | 0.912 | 0.354 |

**Minibatch and parallelism improve speed** These results show that parallel and minibatch variants of the EM algorithm converge substantially faster than the standard method and provide comparable accuracy. With or without parallelism, the minibatch algorithm is more than twice as fast as its counterpart with similar performance. For all of the algorithms, the imputation error of the copula methods improves on that of missForest. This is particularly striking for the online variant, as this algorithm sees each data point only once, suggesting the online algorithm is able to impute surprisingly effectively in only a single pass. Comparing only serial implementations, notice the unthreaded minibatch method is substantially faster than missForest, with substantially lower error for all variable types.

## 4.2. Movie Lens

**Dataset** We also evaluate on the same subset of the MovieLens 1M dataset (Harper & Konstan, 2015), as in Zhao & Udell (2019). This data consists of ordinal ratings on a scale of 1 to 5, with over 75 percent of entries missing.

*Table 2.* Runtimes and Errors on Movie Lens Subset

| Method | Runtime (s) | MAE | RMSE |
|--------|---------|-----|------|
| Standard EM unthreaded | 2411.071 | 0.582 | 0.882 |
| Standard EM threaded | 1033.083 | 0.582 | 0.882 |
| Minibatch EM unthreaded | 446.805 | 0.585 | 0.887 |
| Minibatch EM unthreaded (longer timeout) | 893.929 | 0.583 | 0.884 |
| Online EM threaded | 249.551 | 0.598 | 0.898 |

**Analysis** In the above we can see that the minibatch method obtains comparable performance to the non-minibatch method, and is about 3 times faster, even without parallelism. Comparing to the experiments from Zhao & Udell (2019), we see even the minibatch method has lower MAE than any of the other state-of-the-art methods considered in that work. The error of the online algorithm is appreciably higher than that of the offline methods, but its error is competitive with many of the other state-of-the-art methods (Zhao & Udell, 2019).

### 4.3. Generated data for online setting

**Dataset** Finally, we evaluate the ability of the online algorithm to adapt to a changing covariance structure. To do this we generated a simple dataset from a copula with one continuous, one ordinal, and one binary feature. We use three different correlation matrices in sequence with one set of marginals for all correlations. The correlations are chosen so that their average is approximately the identity. We report mean MAEs over 10 trials and use a batch size of 50 for both the online and offline algorithm. We mask 1 entry at random per row as a test set

*Table 3.* Runtimes and Errors on Simulated Copula Dataset.

| Method | SMAE (Cont) | SMAE (Bin) | SMAE (Ord) | SMAE (Avg) |
|---|---|---|---|---|
| Online EM | 0.840 | 0.907 | 0.763 | 0.836 |
| Offline EM | 0.998 | 1.000 | 1.000 | 0.999 |

**Analysis** In this setting the fully online algorithm outperforms median imputation on average, despite the changing data distribution, by learning to the changing correlation structure online. The offline algorithm, on the other hand, is only able to impute using a single correlation estimate for all of the data; as a result it fails to outperform median imputation. The online algorithm has a sharp spike in error as the covariance abruptly shifts, but the error rapidly declines as the online algorithm learns the new correlation.

## 5. Conclusion and Future Work

In this work we have presented an online semi-parametric algorithm using the Gaussian copula model for mixed, missing value imputation. This model naturally models correlations between binary, ordinal, and continuous variables to impute missing values. We also developed a mini-batch method that is considerably faster than its batch counterpart, with comparable accuracy. We further improved performance by exploiting embarrassing parallelism within minibatches, both in our minibatch method, and directly in the original method. These methods outperform missForest, an existing state of the art method, on both real and synthetic datasets.

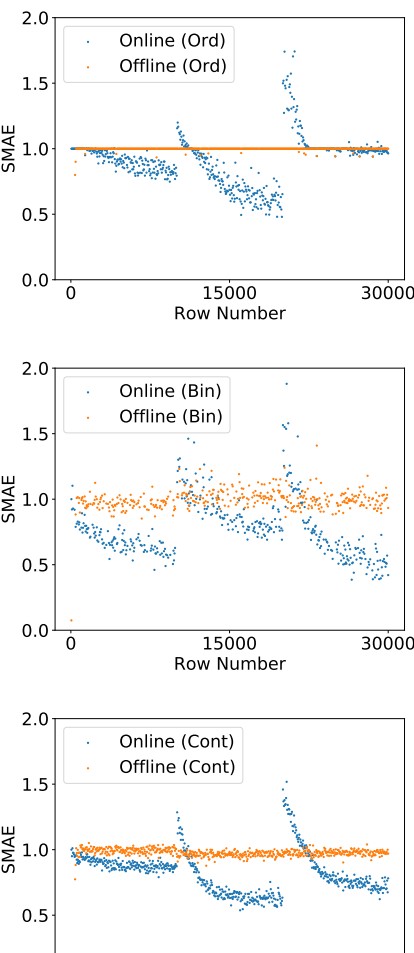

*Figure 1.* Offline vs. online imputation error for ordinal (top), binary (middle) and continuous (bottom) data.

We also provide a proof-of-concept fully-online implementation and that show it can adapt to a changing distribution. One important direction for future work is to extend the method to a streaming data setting by reducing the memory required for marginal estimation. Zhao & Udell (2020) provide a highly performant method for fitting a low-rank Gaussian Copula. Extending this method to the online and streaming setting also constitutes important future work.

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

# A. Supplement for Experiments

## A.1. Generated Data

To generate data from our copula we draw 15-dimension latent vectors i.i.d from a multivariate gaussian with a randomly sampled covariance matrix. We then scale the first five columns to be drawn from an exponential distribution, the next five columns to be binary with a threshold randomly sampled between the 0.1 and 0.9 quantile, and the last five columns to be 5-level ordinals with evenly spaced thresholds. We drawn 2000 elements per copula in each of our evaluations.

## A.2. Movie Lens Subset

The Movie Lens 1M dataset provides one million movie ratings by 6000 users on 4000 movies. We use the subset of 207 movies with at least 1000 ratings and those users who have rated at least 1 of these movies. The resulting set of ratings is over 75% missing.

## A.3. Generated Data for Online Setting

In the online setting we generated data from a copula with 3 features. The first column contains continuous data directly from the latent multivariate Gaussian without scaling. The second column contains ordinal data with 5 levels that are evenly spaces. The final column contains binary data with a random threshold between the 0.1 and 0.9 quantile of the latent values. For each run, we sample 10000 points from each of 3 multivariate gaussians with correlations

$$\Sigma_1 = \begin{bmatrix} 1.0 & 0.339135 & 0.326585 \\ 0.339135 & 1.0 & -0.778398 \\ 0.326585 & -0.778398 & 1.0 \end{bmatrix}$$

$$\Sigma_2 = \begin{bmatrix} 1.0 & -0.778398 & 0.339135 \\ -0.778398 & 1.0 & 0.326585 \\ 0.326585 & 0.326585 & 1.0 \end{bmatrix}$$

$$\Sigma_3 = \begin{bmatrix} 1.0 & 0.326585 & -0.778398 \\ 0.326585 & 1.0 & 0.339135 \\ -0.778398 & 0.339135 & 1.0 \end{bmatrix}$$

We choose these correlations matrices so that their average is approximately the identity. For the online algorithm we use a fixed step size of $\gamma_t = 0.1$. We mask one entry at random per row as a test set. We choose a number of iterations such that the mini-batch algorithm observes each datapoint twice.

# B. Derivations

## B.1. Offline Maximizer

*Proof.*

$$\operatorname*{argmax}_{\Sigma} Q(\Sigma; \Sigma^t, X) = \operatorname*{argmax}_{\Sigma} -\frac{\log(|\Sigma|)}{2}$$

$$-\frac{1}{2}Tr((\Sigma)^{-1}\sum_{i=1}^{n}\frac{1}{n}\mathbb{E}[z^i(z^i)^\top | x^i_{\mathcal{O}_i}, \Sigma^t])$$

Noting that the above is concave with respect to $\Sigma$, and equating the gradient with respect to $\Sigma$ to 0 we get that $\Sigma$ is maximized for

$$0 = \Sigma^{-1} - \Sigma^{-1}\mathbb{E}[z^i(z^i)^\top | x^i_{\mathcal{O}_i}, \Sigma^t]\Sigma^{-1}$$

Rearranging to solve for $\Sigma$ gives us the optimizer

$$\Sigma = \sum_{i=1}^{n}\frac{1}{n}\mathbb{E}[z^i(z^i)^\top | x^i_{\mathcal{O}_i}, \Sigma^t]$$

$\square$

## B.2. Online Maximizer

*Proof.* By unraveling the above recurrence for $Q_t$, it is easy to see that $Q_t(\Sigma) = \sum_{j=1}^{t-1} \alpha_j Q(\Sigma; \Sigma^j, X_S^j)$ for some $\alpha_j \in (0, 1]$ with $\sum_{j=1}^{t-1} \alpha_j = 1$. So we can express the maximization step as

$$\Sigma^{t+1} = \underset{\Sigma}{\operatorname{argmax}} \, Q_{t+1}(\Sigma)$$

$$= \underset{\Sigma}{\operatorname{argmax}} (1 - \gamma_t) \sum_{j=1}^{t-1} \alpha_j Q(\Sigma; \Sigma^j, X_S^j)$$

$$+ \gamma_t Q(\Sigma; \Sigma^t, X_S^t)$$

In what follows we restrict ourselves to considering the case where each $X_S^t$ contains only a single element and denote this by $x_{\mathcal{O}}^i$. The analysis can easily be generalized to larger batches, but we restrict ourselves to this case here for ease of notation. Noting that the above is a convex combination of concave functions, and therefore concave, we equate the gradient of this function with respect to $\Sigma$ to 0 to find its maximizer

$$0 = (1 - \gamma_t) \sum_{j=1}^{t-1} \alpha_j \left( \Sigma^{-1} - \Sigma^{-1} \mathbb{E}[z^j (z^j)^\top | x_{\mathcal{O}_j}^j, \Sigma^j] \Sigma^{-1} \right)$$

$$+ \gamma_t \left( \Sigma^{-1} - \Sigma^{-1} \mathbb{E}[z^t (z^t)^\top | x_{\mathcal{O}_j}^t, \Sigma^t] \Sigma^{-1} \right)$$

$$= (1 - \gamma_t) \sum_{j=1}^{t-1} \alpha_j \left( \Sigma - \mathbb{E}[z^j (z^j)^\top | x_{\mathcal{O}_j}^j, \Sigma^j] \right)$$

$$+ \gamma_t \left( \Sigma - \mathbb{E}[z^t (z^t)^\top | x_{\mathcal{O}_t}^t, \Sigma^t] \right)$$

Rearranging to solve for $\Sigma$

$$\Sigma = \frac{(1 - \gamma_t) \sum_{j=1}^{t-1} \alpha_j \left( \mathbb{E}[z^j (z^j)^\top | x_{\mathcal{O}_j}^j, \Sigma^j] \right)}{(1 - \gamma_t) \sum_{j=1}^{t-1} \alpha_j + \gamma_t}$$

$$+ \frac{\gamma_t \left( \mathbb{E}[z^t (z^t)^\top | x_{\mathcal{O}_t}^t, \Sigma^t] \right)}{(1 - \gamma_t) \sum_{j=1}^{t-1} \alpha_j + \gamma_t}$$

$$= (1 - \gamma_t) \sum_{j=1}^{t-1} \alpha_j \left( \mathbb{E}[z^j (z^j)^\top | x_{\mathcal{O}_j}^j, \Sigma^j] \right)$$

$$+ \gamma_t \left( \mathbb{E}[z^t (z^t)^\top | x_{\mathcal{O}_t}^t, \Sigma^t] \right)$$

$$= (1 - \gamma_t) \Sigma_t + \gamma_t \left( \mathbb{E}[z^t (z^t)^\top | x_{\mathcal{O}_t}^t, \Sigma^t] \right)$$

$\square$