# OpenReview forum: "Online Mixed Missing Value Imputation Using Gaussian Copula"
_ICML.cc/2020/Workshop/Artemiss — ICML Artemiss 2020_

### Official Review · AnonReviewer2 · 2020-06-18
**Online Mixed Missing Value Imputation Using Gaussian Copula**

**Rating:** 9
**Confidence:** 5

**Review:**

The method proposed in this paper is interesting. The abstract is clear.

---

### Decision · Program_Chairs · 2020-07-02

**Decision:**

Accept

**Comment:**

We are very happy to inform you that your paper has been accepted for the Artemiss workshop. We will contact you soon to inform you about the details concerning the format of your presentation at the workshop, and the camera-ready version deadline. Please take into account the referee's comments to write the camera-ready version.